# Long-Term Oncological Outcomes after Nerve-Sparing Robot-Assisted Radical Prostatectomy for High-Risk Localized Prostate Cancer: A Single-Center, Two-Arm Prospective Study

**DOI:** 10.3390/diagnostics14080803

**Published:** 2024-04-11

**Authors:** Lorenzo Spirito, Francesco Chessa, Anna Hagman, Anna Lantz, Giuseppe Celentano, Rodolfo Sanchez-Salas, Roberto La Rocca, Mats Olsson, Olof Akre, Vincenzo Mirone, Peter Wiklund

**Affiliations:** 1Urology Unit, Department of Neurosciences, Reproductive Sciences, and Odontostomatology, University of Naples “Federico II”, 80138 Naples, Italy; lorenzospirito@msn.com (L.S.); roberto.larocca@unina.it (R.L.R.); vincenzo.mirone@unina.it (V.M.); 2Division of Urology, IRCCS Azienda Ospedaliero-Universitaria di Bologna, 40138 Bologna, Italy; 3Department of Urology, Karolinska University Hospital, 171 64 Stockholm, Sweden; anna.hagman@kii.se (A.H.); anna.lantz@kii.se (A.L.); rodolfosanchezsalas@kii.se (R.S.-S.); mats.olsson@kii.se (M.O.); olof.akre@kii.se (O.A.); peter.wiklund@kii.se (P.W.); 4Clinical Epidemiology Unit, Department of Medicine, Karolinska Institutet, 171 77 Stockholm, Sweden; 5Mediterranea Clinic Hospital, 80122 Naples, Italy; dr.giuseppecelentano@gmail.com

**Keywords:** prostate cancer, robot assisted radical prostatectomy, oncologic outcomes, nerve sparing

## Abstract

Aims: To compare the oncological outcomes of patients with high-risk localized prostate cancer undergoing nerve-sparing and non-nerve-sparing robot-assisted radical prostatectomy (RARP). Methods: Between November 2002 and December 2018, we prospectively recorded the data of patients undergoing RARP for high-risk localized prostate cancer (PCa) at our tertiary referral center. NSS (nerve-sparing surgery) was carefully offered on the basis of the preoperative clinical characteristics of the patients and an intraoperative assessment. The patients were stratified into two groups: nerve-sparing and non-nerve-sparing groups (yes/no). Radical prostatectomies were performed by 10 surgeons with a robot-assisted technique using a daVinci^®^ surgical system. The primary oncological outcome evaluated was biochemical recurrence (BCR). The secondary oncological outcomes assessed were positive surgical margins (PSMs) and cancer-specific survival (CSS). Results: A total of 779 patients were included in the study: 429 (55.1%) underwent NSS while 350 (44.9%) underwent non-NSS. After a mean (±SD) follow-up of 192 (±14) months, 328 (42.1%) patients developed BCR; no significant difference was found between the NSS and non-NSS groups (156 vs. 172; *p* = 0.09). Both our univariable and multivariable analyses found that the nerve-sparing approach was not a predictor of BCR (*p* > 0.05). Kaplan–Mayer survival curves for BCR showed no significant difference among the non-NSS, unilateral NSS, and bilateral NSS groups (log rank test = 0.6). PSMs were reported after RARPs for 254 (32.6%) patients, with no significant difference between the NSS and non-NSS group (143 vs. 111; *p* = 0.5). In the subgroup of 15 patients who died during the follow-up period, mean (±SD) CSS was 70.5 (±26.1) months, with no significant difference between the NSS and non-NSS groups (mean CSS: 70.3 vs. 70.7 months). Conclusions: NSS does not appear to negatively impact the oncological outcomes of patients with high-risk PCa. Randomized clinical trials are needed to confirm our promising findings.

## 1. Introduction

Although radical prostatectomy (RP) is the standard of care in cases of localized prostate cancer (PCa), traditionally, surgery may not be preferred over radiotherapy plus androgen deprivation therapy (ADT) in patients with high-risk prostate cancer [1], due to the expected poor oncological and functional outcomes in this population [2]. Even today, about 20–30% of patients are diagnosed with high-risk PCa, showing worse histological characteristics as well as an increased risk of positive surgical margins, biochemical recurrence (BCR), disease progression, and mortality [3].

However, several studies have shown that surgery can be offered in selected high-risk patients, often in the context of multimodal treatment [3]. The feasibility of RARP is proven by the associated oncological and functional outcomes. In fact, approximately 25–30% of patients with high-risk prostate cancer undergoing RARP remain free of biochemical recurrence at 10 years [4]. The preservation of neurovascular bundles can result in better postoperative functional outcomes (in terms of both erectile function and urinary continence) [5]. The robotic approach allows for a better visualization of anatomical structures and a greater precision of movement; thus, nerve-sparing (NS) RARP is capable of yielding an overall potency rate of ≥50%, with an early return of continence in 44–72% of cases [6].

Nerve-sparing surgery (NSS) does not compromise oncological outcomes if patients are carefully selected. According to current European Urology Association (EAU) guidelines, NSS should be performed when there is a low risk of extracapsular extension (based on cT stage, ISUP grade, nomogram, and multiparametric MRI). Therefore, the guidelines do not recommend using NSS in high-risk PCa cases but do not exclude it either.

The aim of this study was to compare the oncological outcomes of patients with high-risk localized prostate cancer undergoing RARP with and without NSS.

## 2. Materials and Methods

### 2.1. Study Design

We designed a single-center, two-arm prospective study at our tertiary referral center (Karolinska Institutet, Solna, Sweden), enrolling patients between November 2002 and December 2018. The oncological outcomes were updated to January 2022. The study was conducted in accordance with the Declaration of Helsinki [7] on ethical principles for medical research involving human subjects, and each patient provided written informed consent to participate. Ethics committee approval was obtained prior to patient enrollment.

### 2.2. Patient Enrollment and Stratification

Consecutive patients with localized high-risk PCa undergoing RARP were included in the study. Subjects who had received prior pelvic radiation, focal therapy for PCa, androgen deprivation therapy (ADT), surgery for benign prostatic hyperplasia, or salvage RARP were excluded from the study.

According to EAU guidelines, high-risk localized PCa was defined on the basis of the following criteria: prostate specific antigen (PSA) score > 20 ng/mL or Gleason score (GS) > 7 or cT2c. Patients with PCa characterized by cT3-4 or cN+ (high risk and locally advanced) were excluded. RARP was offered after appropriate patient counseling and multidisciplinary case discussion. Type of surgery was chosen by the surgeon after a scrupulous evaluation of the preoperative clinical characteristics of patient and disease (as reported in the “Patient assessment” section), as well as of the patient’s intraoperative condition. No nomogram was used to predict the risk of extracapsular extension. Patients were retrospectively stratified into 2 groups according to whether they underwent NSS (yes/no) for the assessment of the oncological outcomes. Some subjects underwent unilateral NSS and were classified as part of the NSS group, but additional analyses were performed to differentiate between unilateral and bilateral NSS groups.

### 2.3. Surgical Details

Radical prostatectomies were performed by 10 surgeons using robot-assisted technique and a daVinci^®^ surgical system (Intuitive Surgical, Sunnyvale, CA, USA). NS dissection was either intra-fascial (dissection in the plane between the prostatic capsule and the periprostatic fascia) or inter-fascial (dissection in the plane between the periprostatic and endopelvic fascia). Non-NS group was divided into patients that underwent either partial non-NS (leaving most of the neurovascular bundle on the prostate) or entirely non-NS. Depending on tumor location, apical dissection was either with maximum urethral length, at the level of the apex, or with a margin from the apex. Bladder-neck-sparing dissection was avoided in patients who were suspected to have bladder neck invasion.

### 2.4. Patient Assessments and Measured Outcomes

All patients underwent a thorough medical history, digital rectal examination (DRE), PSA dosage, transrectal prostate biopsy, and staging imaging before surgery. The histological findings of the biopsies (GS, number, percentage, and location of positive cores) were recorded. Bone scans, abdominal–pelvic computed tomography (CT), and multiparametric magnetic resonance imaging (mpMRI) from 2013 were the imaging techniques used in all patients for staging purposes. Postoperative histological data (pT, pN, postoperative GS, and surgical margin status) and need for adjuvant therapy (radiotherapy or ADT) were reported. Erectile function and urinary continence were evaluated preoperatively to decide whether or not NSS was appropriate. More specifically, the International Index of Erectile Function-Erectile Function (IIEF-EF) test was self-administrated to all patients, and the 24 h pad weight test was used in patients who reported any urine leakage. IIEF-EF score ≤ 16 points or pad weight test score > 100 g/24 h were considered contraindications to NSS.

Primary oncological outcome evaluated was BCR. Secondary oncological outcomes assessed were positive surgical margins (PSMs) and cancer-specific survival (CSS). According to EAU guidelines [8], BCR after radical prostatectomy was defined as PSA score > 0.4 ng/mL and above. Positive surgical margins (PSMs) were defined as the presence of tumor tissue on the inked surface of the specimen [9].

### 2.5. Statistics

The categorical variables are described as frequencies and percentages. The quantitative variables are reported as means and standard deviations (SDs) as well as medians and interquartile ranges (IQRs). The ANOVA (continuous variables) and the Tukey–Kramer test (discrete variables) were used to conduct multiple group comparisons. Kaplan–Meier curve for BCR and multivariate logistic regression analysis to identify main predictors of BCR and CSS were performed. Supportive analysis was performed in order to confirm that the nerve-sparing technique did not influence the risk of biochemical recurrence. Established risk factors, including pre-operatory PSA levels (analyzed as a continuous variable), Gleason score at prostatectomy (analyzed as a dichotomous variable), and pathologic stage (analyzed as a dichotomous variable), were simultaneously assessed using logistic regression as predictors of biochemical recurrence along with the use of nerve-sparing technique. *p*-value threshold was arbitrarily set at 0.05. IBM SPSS Statistics (Released in 2017; IBM SPSS Statistics for Windows, Version 25.0; Armonk, NY, USA: IBM Corp) was used for the statistical analyses.

## 3. Results

A total of 779 patients were included in this study. The mean (±SD) age and PSA score were 63 (±6.4) years and 14 (±14) ng/mL, respectively. The most common biopsy GS was 3+4 (25.7%) (Table 1). Overall, 429 (55.1%) underwent NSS, while 350 (44.9%) underwent non-NSS. Adjuvant therapy was administrated to 49 (11.4%) and 90 (25.7%) patients in the NSS and non-NSS groups, respectively (*p* = 0.07). There was no difference between the NSS and non-NSS groups in terms of pT and pN (*p* > 0.05). Notably, most tumors were ≥ pT3a (non-NSS: 71.1%, NSS: 69.0%), and only a minority of tumors were pN1 (non-NSS: 10.3%, NSS: 7.0%). The PSM was reported after RARP in 254 (32.6%) patients, with no significant difference between the NSS and non-NSS groups (143 vs. 111; *p* = 0.5) (Table 2). The focality (*p* = 0.3), length (*p* = 0.5), and localization (*p* = 0.5) of the PSM were not statistically related to the type of NS approach (Table 3).

After a mean (±SD) follow-up of 192 (±14) months, 328 (42.1%) patients developed BCR. No significant difference was found between the NSS and non-NSS groups in terms of the biochemical recurrence rate (Table 4). Both our univariable and multivariable analyses found that the NS approach was not generally a predictor of BCR (*p* > 0.05), although the bilateral NS group was associated with increased odds of biochemical recurrence compared to the non-NS group (Table 5). The Kaplan–Mayer survival curves for BCR (Figure 1) showed no significant difference among the non-NSS, unilateral NSS, and bilateral NSS groups (log rank test = 0.6) (Figure 1).

The mean (±SD) CSS was 70.5 (±26.1) months. No significant difference was found between the NSS and non-NSS groups (mean CSS: 70.3 vs. 70.7 months) (Table 6).

Both our univariable and multivariable analyses showed that the NS approach was not a predictor of CSS (*p* > 0.05). Our multivariable logistic regression analysis confirmed that the nerve-sparing technique did not influence the risk of biochemical recurrence, even after adjusting for established risk factors such as pathologic stage, Gleason score, and pre-operative PSA levels (Table 7).

## 4. Discussion

Several cohort studies have reported encouraging functional outcomes using an NS approach in patients undergoing RARP [10]. In a retrospective study conducted by Menon et al. [11] and including a total of 1142 prostate cancer patients undergoing NS-RARP with a minimum follow-up period of 12 months, 93% of men with no preoperative erectile dysfunction were able to have intercourse, with an actuarial 5-year BCR rate of 8.4% and a median duration of incontinence of 4 weeks. This cohort included only 8.2% of patients with high-risk prostate cancer. Another smaller, more recent study [12] enrolled 105 prostate cancer patients who underwent intrafascial NS-RARP with a follow-up period of 26.5 months (IQR: 15.25–48). Importantly, they found 6- and 12-month postoperative erectile function recovery rates of 88.6% and 94.3%, with an overall positive surgical margin rate of 16.2% and an overall BCR rate of 6.7%. Also, in this study, patients with high-risk prostate cancer represented a minority of the study sample (18.1%). Conversely, worse sexual recovery rates were reported in a study by Dell’Oglio et al. [13] that included a total of 340 men undergoing Retzius-sparing RARP for high-risk prostate cancer, of whom approximately a third showed BCR during follow-up. In the subgroup of 111 men who were assessable for sexual recovery because they were pre-operatively potent, underwent full or partial NSS, and were adequately followed-up with, only 43.1 and 50% achieved sexual recovery at 1 and 2 years, respectively. The lower sexual recovery rate reported by Dell’Oglio et al. may be due to the substantial proportion of patients who underwent unilateral NS, which is less effective compared to bilateral NS in preserving potency [14]. Also, NS-RARP seems to yield improved functional outcomes compared to NS laparoscopic radical prostatectomy (LRP). Asimakopoulos et al. [15] assessed pentafecta outcomes associated with RARP vs. LRP using a bilateral NS approach and combined oncological and functional outcomes, along with an assessment of surgical margins and evaluation of early post-surgical complications. Importantly, the authors reported that 45.6% vs. 27.5% of the patients achieved pentafecta outcomes following RARP vs. LRP. Ou et al. reported a pentafecta rate of 60.4% in a prospective cohort of 230 prostate cancer patients undergoing bilateral NS-RARP [16], while Jazayeri et al. reported a pentafecta rate of 73.9% in a retrospective study of 566 prostate cancer patients treated with NS-RARP [17]. Overall, these summarized findings suggest that NS-RARP has a tremendous potential for improved functional outcomes, while its potentially negative implications in terms of oncologic outcomes need to be further investigated, especially in patients with a higher risk of recurrence.

In our prospective cohort of men receiving NS- and non-NS-RARP, who were followed up for a mean time > 10 years, we focused on oncological outcomes only and proved that NSS was not associated with any detrimental effect in terms of the positive margin rate, BCR rate, and CSS. Of note, these results were achieved in a population of men affected by high-risk prostate cancer, with approximately 70% of these patients showing ≥ pT3a disease, which supports the notion that an NS approach can be adopted in selected men with high-risk PCa.

Our study has several strengths, which include a large sample size, a long follow-up period, and its prospective design. On the other hand, our research presents multiple limitations. Although a comparison between patients receiving NS- and non-NS-RARP was our main objective, the study design did not include any sample size calculations to assess the non-inferiority of the techniques between them. In this regard, it must be noted that the BCR rate was numerically (albeit not significantly) superior in the NS- vs. non-NS-RARP patients. We are uncertain whether a larger sample size may have allowed this to reach statistically significance. Furthermore, the patients were not randomized to the NS- vs. non-NS-RARP groups, so we cannot exclude that the two groups were actually unbalanced for some clinically meaningful variable that we could not account for, and patients with a lower risk of recurrence may have been more likely to undergo NS-RARP. Importantly, in our study, we observed that the biochemical recurrence (BCR) rate was numerically higher in patients undergoing nerve-sparing robot-assisted radical prostatectomy (NS-RARP), although this difference did not reach statistical significance. It is essential to acknowledge that with a larger sample size, there might be a potential to detect statistically significant differences in BCR rates between NS-RARP and non-NS-RARP patients.

Also, the lack of randomization between the NS-RARP and non-NS-RARP patients introduces concerns regarding potential bias and confounding variables. Without random allocation, it is challenging to determine whether the observed outcome differences are due to the surgical technique itself or other underlying patient characteristics. Additionally, the possibility of unaccounted variables leading to group imbalances underscores the need for a cautious interpretation of our study’s findings.

In conclusion, our large prospective study spanning over 10 years aimed to investigate potential differences in oncological outcomes among high-risk prostate cancer (PCa) patients undergoing nerve-sparing (NS) versus non-nerve-sparing (non-NS) robot-assisted radical prostatectomy (RARP). We found no statistically significant difference in the positive margin rate, biochemical recurrence (BCR), or cancer-specific survival (CSS) between the two groups. Both our univariable and multivariable analyses indicated that the NS approach was not generally predictive of BCR, although the bilateral NS approach was associated with increased odds of biochemical recurrence compared to the non-NS approach. Our findings provide preliminary evidence supporting the use of an NS approach in high-risk PCa patients undergoing RARP. Nevertheless, further well-designed randomized controlled trials (RCTs) are warranted to validate and expand upon our promising results.

## Figures and Tables

**Figure 1 diagnostics-14-00803-f001:**
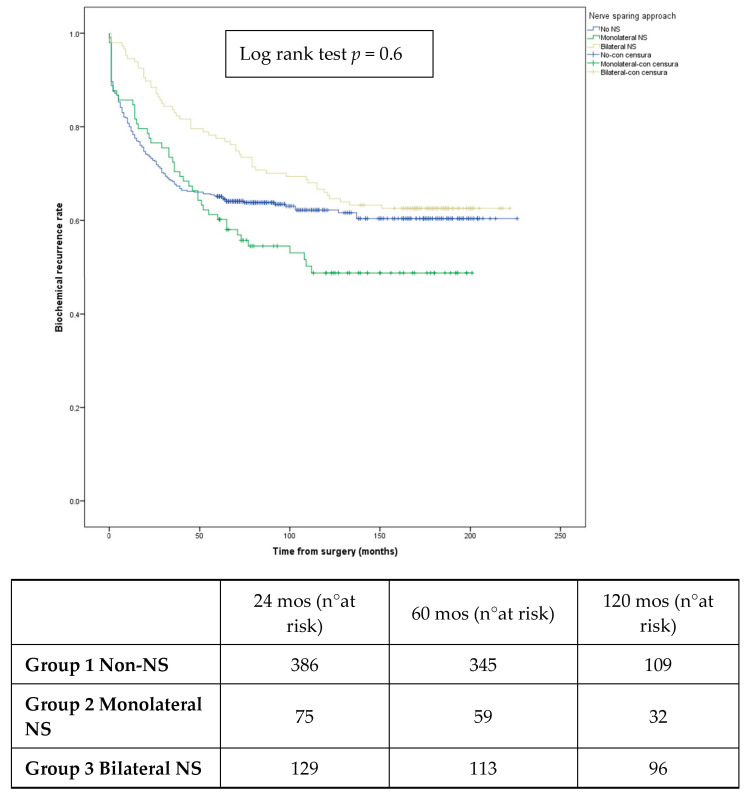
Kaplan–Meier survival curves for biochemical recurrence rate stratified by nerve-sparing approach.

**Table 1 diagnostics-14-00803-t001:** Demographic and clinical characteristics.

Variable	Overall
Age (years)	
Mean (±SD)	63 (±6.4)
Median (IQR)	64 (59–68)
PSA (ng/mL)	
Mean (± SD)	14 (±14)
Median (IQR)	8.90 (5–19)
Nerve Sparing, *n* (%)	
Yes	429 (54.8%)
No	350 (45.2%)
Clinical Gleason Score, *n* (%)	
3+3	113 (14.56%)
3+4	200 (25.71%)
4+3	84 (10.79%)
4+4	191 (24.64%)
>4+4	188 (24.28%)

**Table 2 diagnostics-14-00803-t002:** Postoperative pathologic characteristics.

Variable	Overall	Group 1 (*N* = 350) non ns	Group 2 (*N* = 429) ns	*p* Value
Pathologic stage, *n* (%)				0.08
pT2	pT2: 230 (29.56%)	pT2: 101 (28.85%)	pT2: 129 (30.14%)	
pT3a	pT3a: 393 (50.51%)	pT3a: 149 (42.57%)	pT3a: 244 (57%)	
pT3b	pT3b: 145 (19.02%)	pT3b: 96 (27.42%)	pT3b: 49 (12.15%)	
pT4	pT4: 7 (0.9%)	pT4: 4 (1.14%)	pT4: 3 (0.7%)	
Lymph node status, *n* (%)				0.09
pN0	pNx: 360 (46.43%)	pNx: 149 (42.69%)	pNx: 211 (49.42%)	
pNx	pN0: 349 (44.91%)	pN0: 164 (47%)	pN0: 185 (43.24%)	
pN1	pN1: 66 (8.65%)	pN1: 36 (10.31%)	pN1: 30 (7.32%)	
Pathologic Gleason score, *n* (%)				0.02
3+3	3+3: 81 (10.51%)	3+3: 23 (6.68%)	3+3: 58 (13.61%)	
3+4	3+4: 218 (27.66%)	3+4: 75 (21.80%)	3+4: 143 (32.39%)	
4+3	4+3: 203 (26.36%)	4+3:88 (25.58%)	4+3:115(26.99%)	
4+4	4+4: 137 (17.79%)	4+4: 65 (18.89%)	4+4: 72(16.90%)	
>4+4	>4+4: 136 (17.66%)	>4+4: 93 (27.03%)	>4+4: 43(10.09%)	
Positive surgical margins, *n* (%)	254 (32%)	111 (31%)	143 (33%)	0.5

**Table 3 diagnostics-14-00803-t003:** Detailed reports of margin status according to nerve-sparing approach.

Variable	Overall	Group 1 (*N* = 350) non-ns	Group 2 (*N* = 429) ns	*p* Value
Negative surgical margins, *n* (%)	524	239	282	0.3
Monofocal PSM	162	65	96	
Multifocal PSM	93	46	47	
Positive surgical margins length, *n* (%)				0.5
≤3 mm	130	53	77	
>3 mm	123	57	66	
Localization of the PSM, *n* (%)				0.5
Base	63	28 (8%)	30 (7%)	
Posterolateral	100	35 (10%)	65 (15%)	
Anterior	54	27 (8%)	27 (6%)	
Apex	91	41 (12%)	50 (12%)	

**Table 4 diagnostics-14-00803-t004:** Oncologic outcomes of patients, stratified by nerve-sparing surgery.

Variable	Overall	Group 1 (*N* = 350)	Group 2 (*N* = 429)	*p* Value
Follow-up (months)				0.7
Mean (±SD)	192 (±14)	219 (±448)	170 (±296)	
Median (IQR)	132 (80–180)	133 (85–181)	125 (78–180)	
Biochemical recurrence rate, *n* (%)	No: 447 (58.33%) Yes: 328 (41.67%)	No: 178 (54.57%) Yes: 172 (45.43%)	No: 273 (61.39%) Yes: 156 (38.61%)	0.09
Adjuvant therapy, *n* (%)	Yes: 135 (17.67%) No: 640 (82.33%)	Yes: 90 (25.14%) No: 260 (74.86%)	Yes: 49 (15.10%) No: 380 (84.90%)	0.07

**Table 5 diagnostics-14-00803-t005:** Univariable and multivariable analyses for predictors of biochemical recurrence.

**Univariable Analysis for Predictors of Biochemical Recurrence Rate**				
**Variable**	**OR**	**Lower CI**	**Upper CI**	***p* Value**
Age > 70 years	1	0.7	1.5	0.7
Preoperative PSA value				<0.001
≤10 ng/mL vs. >10 ng/mL	1.6	1.2	2.0	
Positive surgical margin (PSM)	1.5	1.2	1.9	<0.001
Surgical margin status				<0.001
Negative	Reference			
Positive Monofocal	1.3	1.1	1.9	0.005
Positive Multifocal	1.6	1.2	2.1	<0.001
PSM length				0.007
<3 mm vs. ≥3 mm	1.4	1.1	1.9	
Pathologic Gleason score				<0.001
3+3	Reference			
≤3+4 vs. 3+3	1.6	1	2.8	0.05
4+3 vs. 3+3	2.3	1.3	3.8	0.01
4+4 vs. ≤ 3+3	2.6	1.5	4.5	0.001
≥4+5 vs. ≤3+3	3.8	2.3	6.4	0.001
Pathologic stage				
Organ confined vs. Locally advanced	2.0	1.5	2.7	<0.001
Lymph node status				<0.001
Nx	Reference			
N0 vs. Nx	0.9	0.7	1.1	0.5
N1 vs. Nx	4.3	3.0	6.0	<0.001
Nerve-sparing (NS) approach				0.6
Non-NS	Reference			
Monolateral NS vs. Non-NS	1.2	0.9	1.7	0.1
Bilateral NS vs. Non-NS	0.7	0.5	1.0	0.1
**Multivariable Analysis for Predictors of Biochemical Recurrence Rate**				
**Variable**	**OR**	**Lower CI**	**Upper CI**	***p* Value**
Age > 70 years	1.0	0.9	1.5	0.6
Preoperative PSA value				<0.001
≤10 ng/mL vs. >10 ng/mL	1.4	1.5	2.8	
Positive surgical margin (PSM)	1.5	1.3	1.8	0.005
Surgical margin status				
Negative	Reference			<0.001
Positive Monofocal	1.4	0.9	2.4	0.1
Positive Multifocal	2.3	1.5	3.6	<0.001
PSM length				
<3 mm vs. ≥3 mm	1.5	0.8	2.8	0.1
Pathologic Gleason score				
3+3	Reference			<0.001
≤3+4 vs. 3+3	1.6	0.9	2	0.07
4+3 vs. 3+3	2.1	1.2	3.5	<0.001
4+4 vs. ≤3+3	2.5	1.4	4.4	<0.001
≥4+5 vs. ≤3+3	3.4	1.9	6	<0.001
Pathologic stage				
Organ confined vs. Locally advanced	2.1	1.5	2.1	<0.001
Lymph node status				
Nx	Reference			<0.001
N0 vs. Nx	0.7	0.7	1	0.1
N1 vs. Nx	2.5	1.6	3.8	<0.001
Nerve-sparing (NS) approach				
Non-NS	Reference			0.06
Monolateral NS vs. Non-NS	0.7	0.5	1.5	0.7
Bilateral NS vs. Non-NS	1.3	1.0	1.8	0.03

**Table 6 diagnostics-14-00803-t006:** Univariable and multivariable analyses for predictors of cancer-specific survival.

**Univariable Analysis for Predictors of Cancer-Specific Survival**				
**Variable**	**HR**	**95% CI**	**Upper CI**	***p* Value**
Age > 70 years	0.3	0.04	2.8	0.3
Preoperative PSA value				
≤10 ng/mL vs. >10 ng/mL	0.5	0.2	1.6	0.3
Positive surgical margin (PSM)	1.1	0.4	3.1	0.7
Surgical margin status				
Negative	Reference			0.7
Positive Monofocal	1.4	0.4	4.5	0.5
Positive Multifocal	0.8	0.2	3.2	0.8
PSM length				
<3 mm vs. ≥3 mm	0.7	0.2	2.3	0.6
Pathologic Gleason score				
3+3	Reference			0.03
≤3+4 vs. 3+3	0.8	0.1	5.0	0.8
4+3 vs. 3+3	0.7	0.1	11	0.7
4+4 vs. ≤3+3	1.4	0.1	4.0	0.7
≥4+5 vs. ≤3+3	16.2	1.7	150	0.01
Pathologic stage				0.2
Organ confined vs. Locally advanced	3.3	0.4	26	
Lymph node status				
Nx	Reference			0.04
N0 vs. Nx	1.4	0.5	4.1	0.6
N1 vs. Nx	1.5	1.5	5.9	<0.001
Nerve-sparing (NS) approach				
Non-NS	Reference			0.09
Monolateral NS vs. Non-NS	0.9	0.9	1.5	0.09
Bilateral NS vs. Non-NS	1.0	0.9	1.8	0.09
**Multivariable Analysis for Predictors of Cancer-Specific Survival**				
**Variable**	**HR**	**Lower CI**	**Upper CI**	***p* Value**
Age > 70 years	0.6	0.2	20	0.8
Preoperative PSA value				
≤10 ng/mL vs. >10 ng/mL	0.1	0.01	1.6	0.1
Positive surgical margin (PSM)	0.4	0.3	11	0.8
Surgical margin status				
Negative	Reference			0.9
Positive Monofocal	0.9	0.7	1.8	0.8
Positive Multifocal	1.0	0.04	90	0.9
PSM length				
<3 mm vs. ≥3 mm	0.9	0.05	16	0.9
Pathologic Gleason score				
3+3	Reference			0.2
≤3+4 vs. 3+3	0.6	0.04	11	0.8
4+3 vs. 3+3	16	0.6	456	0.09
4+4 vs. ≤3+3	0.7	0.03	20	0.9
≥4+5 vs. ≤3+3	0.2	0.01	4	0.3
Pathologic stage				0.1
Organ confined vs. Locally advanced	11	0.5	243	
Lymph node status				
Nx	Reference			0.001
N0 vs. Nx	1.2	0.8	1.5	0.06
N1 vs. Nx	41	1.6	1000	<0.001
Nerve-sparing (NS) approach				
Non-NS	Reference			0.09
Monolateral NS vs. Non-NS	0.9	0.9	1.5	0.09
Bilateral NS vs. Non-NS	1.0	0.9	1.8	0.09

**Table 7 diagnostics-14-00803-t007:** Multivariable analysis including nerve-sparing technique and others factors of established predictive value.

Variable	Odds Ratio for Biochemical Recurrence	95% Confidence Interval	*p*
Nerve-sparing technique (Yes vs. no)	0.89	0.65 to 1.23	0.5064
Pathologic stage (pT3-4 vs. pT1-2)	2.72	1.90 to 3.90	<0.0001
Preoperative PSA levels (continuous variable)	1.02	1.01 to 1.03	0.0006
Gleason score (8–10 vs. 6–7)	1.89	1.36 to 2.62	0.0001

## Data Availability

Data are contained within the article.

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
