# Peer review of "Long-Term Oncological Outcomes after Nerve-Sparing Robot-Assisted Radical Prostatectomy for High-Risk Localized Prostate Cancer: A Single-Center, Two-Arm Prospective Study"

_diagnostics, 2024, doi:10.3390/diagnostics14080803_

Round 1

Reviewer 1 Report

Comments and Suggestions for Authors

A well-designed original study that aims to investigate the potential of NS surgery in high risk PCa patient's.

The article is well written and the analyzes well reported.

The study is powered by a large number of randomized patients and long follow-up. The limitations were well mentioned.

Conclusions are consistent with the evidence presented and addressed the main question posed.

Line 129: Correct bioptic → biopsy

Figures and tables are ok.

Congratulations for the work in this paper.

Author Response

Thank you so much for the revision

Reviewer 2 Report

Comments and Suggestions for Authors

The manuscript presents several strengths in terms of its large sample size, long follow-up period, and prospective design, which contribute to its scientific rigor. However, the study also exhibits notable limitations that warrant consideration.

Firstly, the absence of a perioperative outcome calculation to assess non-inferiority between the NS- and non-NS- RARP techniques raises concerns regarding the robustness of the study's conclusions. Given the numerical but non-significant superiority in biochemical recurrence rate observed in NS-RARP patients, the potential impact of a larger sample size on achieving statistical significance merits acknowledgment. Thus, it is necessary to analyze some perioperative outcomes.

Additionally, the lack of randomization between patients receiving NS- and non-NS- RARP raises questions about the potential for bias and confounding variables. Without random allocation, it remains unclear whether the observed differences in outcomes between the two groups are truly attributable to the surgical technique or to underlying patient characteristics. Appropriate optimization of statistical methods such as weighted matching or PSM to balance the two sets of baseline data. 

Furthermore, the possibility of unaccounted variables leading to group imbalances introduces uncertainty into the interpretation of the study findings. Without controlling for potentially confounding factors, such as patients' risk profiles, the validity of the comparisons made between NS- and non-NS- RARP patients may be compromised.

Author Response

the limitations highlighted in the review are indeed typical of retrospective studies and are inherent to the nature of our investigations. We recognize the need for a prospective randomized trial to definitively address the questions raised.,

Pur manuscript primari serves to lay the groundwork and construct rationale for such a prospective trial.

While we acknowledge the limitations mentioned, we believe that our study provides valuable insights despite these constraints. We have taken the reviewer's feedback into careful consideration and have been more meticulous in our conclusion, acknowledging the limitations and emphasizing the need for further research to confirm and expand upon our findings

Reviewer 3 Report

Comments and Suggestions for Authors

Lines 156-158: In this multivariate model, were Unilateral and Bilateral NS both found to nonsignificantly impact BCR? Only one Odds Ratio is reported so it seems as though the authors aggregated bilateral and unilateral NS. Is this correct? This should be clarified. Additionally, why did the authors perform this secondary multivariate analysis? It seems that this model includes less covariates than the model shown in Table 3. 

Lines 158-163: It is interesting to see that in these surgical margin subgroups, NS approach did not significantly impact BCR. However, it is unclear whether the authors used unilateral NS, bilateral NS, or any NS as the comparator to No-NS. 

Lines 200-203: Did the authors assess for post-operative erectile function rates with unilateral and bilateral NS? This study is novel in its application of NS for high-risk PCa, and it would be interesting to also see the efficacy of NS in achieving favorable postoperative functional outcomes in this scenario. 

Lines 205-207: Claiming that these results “prove” that NSS is not associated with BCR rates contradicts the finding that bilateral NSS was significantly associated with BCR (Table 3). Furthermore, the absence of a power analysis leads this study suspect to potential Type 2 error, especially when comparing unilateral NS and no-NS. While the absence of a-priori sample size calculation is mentioned in the limitations section, the authors should adjust their conclusions to consider these limitations. The authors could also consider performing an ad-hoc power analysis in lieu of the absence of post-priori sample size calculation.

Lines 224-225: Once again, the authors conclusion that there was no statistically significant difference in BCR with NS contradicts the positive finding with bilateral NS highlighted in Table 3. 

Author Response

Table 3.

In an attempt to provide further exploratory analysis and increase statistical power, we explored the impact of nerve sparing on biochemical recurrence by considering only the D'Amico Classification variable, including nerve-sparing technique (Yes vs. No), pathologic stage (pT3-4 vs. pT1-2), preoperative PSA levels (continuous variable), and Gleason score (8-10 vs. 6-7), while unifying unilateral and bilateral nerve sparing. The decision to perform this secondary multivariate analysis was driven by the aim to investigate the specific impact of nerve sparing on biochemical recurrence, while simplifying the model to focus on key variables of interest.Regarding the aggregation of bilateral and unilateral nerve sparing, this approach was taken to ensure an adequate sample size for analysis and to reduce potential confounding effects. However, it's important to note that the distinction between unilateral and bilateral nerve sparing may still hold clinical significance and should be considered in future studies.Furthermore, while our study provides valuable preliminary data, we acknowledge the need for prospective comparative trials to definitively clarify whether nerve-sparing surgical techniques have comparable clinical outcomes. Our analysis serves as a foundation for informing the design and rationale of such prospective trials.

Reviewer 4 Report

Comments and Suggestions for Authors

Congratulations on the present manuscript!

Your work is highly documented, and the data are presented in a well-designed way.

Even though there are some aspects that may improve the paper's quality.

1. In line 72 the authors should include a citation of the EAU Guideline.

2. You included 10 surgeons in the present study. What was their experience with daVinci system? It should be specified. 

3. The discussion section should be extended and may include more data comparisons.

4. Please try to avoid paragraphs starting with words like "we designed, "and "our perspective." These expressions should be replaced with more scientific words without personal involvement. 

Author Response

Thank you so much for your suggestions.

We will include the EAU Guideline citations in the line 72.

The 10 surgeons included in the study have wide experience with robotic surgeons and all of them have overcome the learning curve for RARP.

We will extend the discussion

Round 2

Reviewer 3 Report

Comments and Suggestions for Authors

The results reported in the Multivariate analysis for prediction of biochemical recurrence rate and the Kaplan-Meier plot for biochemical recurrence rate seem to contradict each other. Specifically, the multivariate regression found that bilateral NS significantly increased odds of BCR, contradicting the KM that shows that there was no significant difference in BCR between bilateral NS and No-NS. How do the authors reconcile these findings?

Table 5 - It is unclear why some p-values are bolded. The authors should only bold p-values under 0.05 for clarity.

Author Response

The discrepancy between the findings of the multivariate analysis and the Kaplan meier (km) plot regarding the impact of bilateral nerve sparing (NS) on biochemical recurrence (BCR) can be reconciled by considering the nature of each analysis.

The multivariate analysis, being a comprehensive assessment that considers multiple variables simultaneously, indetified bilateral NS as significanlty incrasing the odds of BCR. On the other hand, the KM plot, wich represents a univariate analysis focusing solely on survival utcomes over time, did not show a significant difference in BCR between NS and No-NS.

This inconsistency arises because the KM plot does not account for the influence of multiple variables, whereas the multivariate analysis does. 

Therefore the apparent contradiction can be resolved by understanding that the KM lot is consistent with a univariate analysis approach and may not fully capture the impact of bilateral NSS when considering other covariate.

We corrected p values inappropiately in bold